# Spatially selective cell treatment and collection for integrative drug testing using hydrodynamic flow focusing and shifting

Xu Wang[1], Jingtian Zheng[1], Maheshwar Adiraj Iyer[2], Adam Henry Szmelter[2], David T. Eddington[2,3]*, Steve Seung-Young Lee[1,3]*

**1** Department of Pharmaceutical Sciences, University of Illinois Chicago, Chicago, Illinois, United States of America, **2** Department of Biomedical Engineering, University of Illinois Chicago, Chicago, Illinois, United States of America, **3** University of Illinois Cancer Center, University of Illinois Chicago, Chicago, Illinois, United States of America

* dte@uic.edu (DTE); ssylee@uic.edu (SSYL)

**Data Availability Statement:** All relevant data are within the paper and its Supporting Information files.

## Abstract

Hydrodynamic focusing capable of readily producing and controlling laminar flow facilitates drug treatment of cells in existing microfluidic culture devices. However, to expand applications of such devices to multiparameter drug testing, critical limitations in current hydrodynamic focusing microfluidics must be addressed. Here we describe hydrodynamic focusing and shifting as an advanced microfluidics tool for spatially selective drug delivery and integrative cell-based drug testing. We designed and fabricated a co-flow focusing, three-channel microfluidic device with a wide cell culture chamber. By controlling inlet flow rates of sample and two side solutions, we could generate hydrodynamic focusing and shifting that mediated precise regulation of the path and width of reagent and drug stream in the microfluidic device. We successfully validated a hydrodynamic focusing and shifting approach for spatially selective delivery of DiI, a lipophilic fluorophore, and doxorubicin, a chemotherapeutic agent, to tumor cells in our device. Moreover, subsequent flowing of a trypsin EDTA solution over the cells that were exposed to doxorubicin flow allowed us to selectively collect the treated cells. Our approach enabled downstream high-resolution microscopy of the cell suspension to confirm the nuclear delivery of doxorubicin into the tumor cells. In the device, we could also evaluate *in situ* the cytotoxic effect of doxorubicin to the tumor cells that were selectively treated by hydrodynamic flow focusing and shifting. These results show that hydrodynamic focusing and shifting enable a fast and robust approach to spatially treat and then collect cells in an optimized microfluidic device, offering an integrative assay tool for efficient drug screening and discovery.

## Introduction

Microfluidics have advanced immensely over the past decades, becoming an indispensable technology in biomedical and pharmaceutical research [1, 2]. A microfluidic device capable of adequately housing live cells and precisely controlling microliter volume fluids offers a

**Funding:** This study was supported by NIBIB (National Institute of Biomedical Imaging and Bioengineering) R00 EB0022636 and NIGMS (National Institute of General Medical Sciences) R35 GM142743 to S. S.-Y. Lee. Experiments were performed in a renovated laboratory space at University of Illinois Department of Pharmaceutical Sciences, supported by NIH (National Institutes of Health) C06 RR015482. The funders had no role in study design, data collection and analysis, decision to publish, or preparation of the manuscript.

**Competing interests:** The authors have declared that no competing interests exist.

miniaturized *in vitro* platform for cell biology and cell-based drug assays [3, 4]. Highly predictive preclinical drug testing is essential to increasing success rates in drug development [5]. Although incubation of cells with drug candidates in a multiwell format is the gold standard for high-throughput drug testing *in vitro*, static treatment conditions do not emulate the exposure of cells to drug molecules *in vivo*. In contrast, microfluidics systems can generate a controlled stream of drug solutions through microchannels where cells are located. This dynamic condition mimics an *in vivo* model with a tissue environment where cells interact and take up drug molecules being circulated in the bloodstream after administration [6, 7]. Importantly, microfluidics can produce laminar flow having a width of less than 50 nm [8]. These technical advantages enable spatial and temporal regulation of the cellular environment and intracellular biochemical processes. Such regulation could be possible through selective spatiotemporal exposure of a given part of a cell with a reagent stream (e.g. chemical stimulant or perturbation), which facilitates the studies of subcellular or organelle biology [9–11]. Cell patterning in a microfluidic device for studying cell migration and cell-cell interactions in the context of a wound healing assay is also possible with laminar flow [12, 13]. These cell analysis platforms can be viable by hydrodynamic flow focusing in a relatively simple microfluidics device [14, 15].

Previous microfluidics studies using hydrodynamic flow focusing have shown promising results in cell-based drug testing [16, 17]. Wang and co-workers have designed a hydrodynamic flow focusing microfluidic device with a cross-shaped channel connecting three input ports (two sides for buffer and one central for sample) and one output port [18]. By changing the flow rate ratio of the side and sample fluids, the authors could control the focus width of the sample stream and selectively deliver a model drug molecule to CHO cells located along the central channel (100 μm width). A two-layer microfluidic system consisting of a circle-shaped cell culture chamber (1.2 mm in diameter) and channels in different layers has also been introduced to expose cells to hydrodynamically focused streams of soluble bioreagents under low shear stress [19]. Another multi-layer and -channel microfluidic device has been developed for a long-term cell study in which myoblasts differentiated and formed a microtube pattern in the cell trench (L/W/H = 400/80/110 μm or 360/60/110 μm) [20]. The device generated a hydrodynamically focused reagent stream and enabled focal delivery of cytoplasmic and nuclear dyes to micro-patterned myoblasts and myofibers. In addition, hydrodynamic flow focusing facilitated a cell migration study by generating different physiological environments in a microfluidics device with a 100 μm-wide culture channel [21]. The microfluidic device was composed of separate cell culture and chemical reaction channels that produce spatially defined oxygen gradients and cell patterns, enabling characterization of oxygen level-dependent endothelial cell migration.

These pioneering efforts demonstrate the potential of hydrodynamic flow focusing for spatially selective drug delivery to cells and testing the effects of drugs on cells. However, most of the previously reported hydrodynamic flow focusing microfluidic devices have been designed to have a small cell culture space (~0.1–1 mm) and to deliver a drug or a bio-molecule to cells within a confined area, typically the centerline of the main channel. Although the devices facilitate response monitoring of a few targeted cells after direct treatment with a single drug, these approaches make it difficult to study the effects of different drug molecules and concentrations on cells in a device. Thus, currently used approaches require much time and cost to prepare many devices and culture cells for testing various drugs. A microfluidics system capable of culturing cells in a large chamber and shifting the position of a hydrodynamically focused flow of drug solution has not yet been reported for simultaneous evaluation of cell responses to various drug treatments in the same device. Furthermore, integration of hydrodynamic focusing microfluidics with other drug assay methods, such as high-resolution microscopy and single-

cell sequencing, has also not yet been assessed, which would provide further detailed information of drug effects on individual cells.

Here we introduce a novel hydrodynamic flow focusing and shifting approach for spatially selective drug delivery and cell collection in a simple microfluidic culture device. We designed three different inlet geometries and characterized them by computational simulation to determine an optimal device design for efficiently generating a stable laminar flow and controlling its position. Then, we fabricated a novel hydrodynamic focusing and shifting microfluidics (HFSM) device with a sample/side co-flow input channel and a large cell culture chamber (2 mm width). The device provided an appropriate environment for cancer cells to proliferate and grow, and successfully cultured millions of cells after 7 days of incubation. We generated hydrodynamic flow focusing and shifting by controlling inlet flow rates of sample and side fluids. Lateral shifting of the central stream could be made within a range of 1 mm, which was confirmed by both 2D laminar flow simulations and a real experiment using a food coloring dye. We applied the hydrodynamic focusing and shifting for spatially controlled delivery of DiI, a lipophilic fluorescent dye, and doxorubicin, a chemotherapy drug, to cancer cells localized in different regions of the culture chamber. To evaluate the cytotoxic effect of doxorubicin *in situ*, we monitored the drug-treated cells using a brightfield microscope with a low magnification objective and then observed spatially selective cell death. Direct cell exposure to the drug in the perfusion stream caused fast delivery and cellular response. We also selectively collected the cells exposed to doxorubicin flow by precisely controlling the stream path of a trypsin EDTA solution. Then, we applied high-resolution microscopy for imaging the single cells to confirm nuclear delivery of doxorubicin where it acts pharmacologically. These results show great potential of hydrodynamic focusing and shifting as a novel microfluidic tool for integrative cell-based drug testing.

## Materials and methods

### Cell culture

The cell line used in this study was mouse mammary gland tumor cells (4T1 cells) obtained from Dr. Stephen J. Kron's group at the University of Chicago. 4T1 cells were cultured in RPMI-1640 media (Gibco, U.S.A.) containing 10% fetal bovine serum (FBS) and 1% Penicillin-Streptomycin (PS). Cells were maintained at 37˚C with 5% $CO_2$ in a humidified atmosphere.

### Hydrodynamic focusing and shifting microfluidics (HFSM) design and fabrication

We tested three different HFSM designs having three different inlet geometries: 1) cross-channel with an inverted T-shape junction, 2) 45˚ tilted side channel with a down arrow-shaped junction, and 3) sample/side co-flow channel without a junction. We evaluated and determined an optimal HFSM device design using 2D laminar flow simulation. We also simulated the time-dependent concentration change of a shifted sample stream in the selected design. The design of the optimized HFSM device is illustrated in Fig 1A. The height of all the channels are 160 μm. Co-flow of side fluids hydrodynamically squeeze the center sample fluid to generate a laminar stream and can also shift the sample stream by uneven inlet flow ratios. The culture chamber is located underneath the inlet channels where cells are loaded and grown. The widths of the inlet channels $W_{in}$ and cell culture chamber $W_{cc}$ were designed to be 200 μm and 2 mm, respectively.

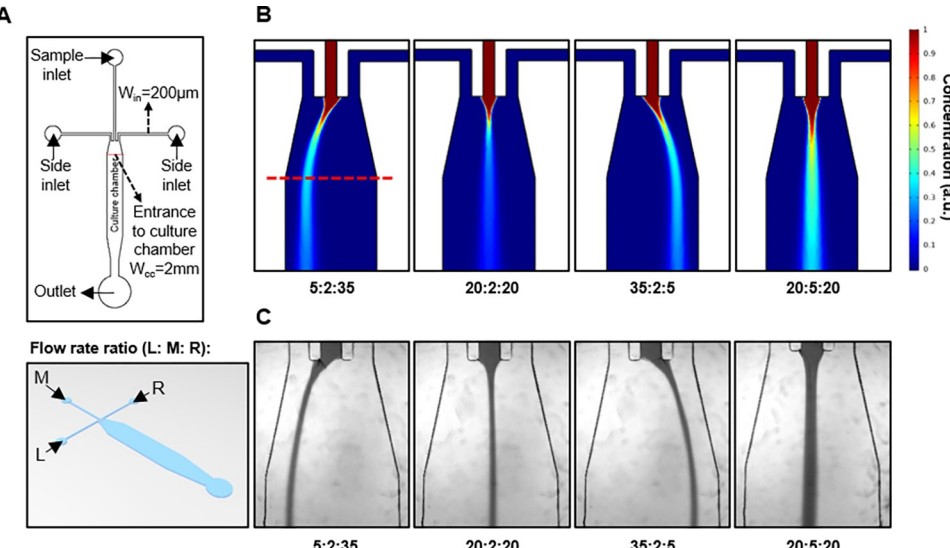

**Fig 1. Design, simulation and experimental test of hydrodynamic focusing and shifting microfluidics (HFSM). A** Design of HFSM device. **B** Simulation and **C** experimental testing of sample fluid in the HFSM device using a brightfield microscope and food dye solution. The red dotted lines in **A** and **B (left)** indicate the entrance to the cell culture chamber.

The HFSM device was fabricated from polydimethylsiloxane (PDMS, Sylgard 184, Dow Corning) using replica molding methods. The master mold was fabricated by patterning a layer of photoresist (SU-8, Kayaku Advanced Materials) through photolithography. PDMS elastomer and curing agent were mixed in a ratio of 10:1 (w/w) and degassed before pouring onto the master mold. After curing on a hot plate at 95°C for 1 hour, the PDMS was cross-linked and peeled off from the master mold. Then, inlet and outlet ports were made with a 1.0 mm biopsy puncher and the PDMS microfluidic network (2 mm thick) was bonded to a 1 mm-thick glass slide using oxygen plasma. The bonded microfluidic device was cleaned by flowing isopropyl alcohol and deionized water through all the channels. Finally, the fabricated microfluidic device was autoclaved and stored in a sterilized biosafety cabinet before cell loading.

## Loading cells into the culture chamber of the HFSM

Before loading cells, the channels of the HFSM were pre-incubated with the cell culture media for 30 min. 4T1 cells were prepared in a T-75 flask and harvested in cell culture media at a concentration of 500,000 cells/mL. A cell solution of 100 μL was injected into the culture chamber of the HFSM through the center channel using a 1 mL syringe (BD Luer-Lok™, sterile) and 0.02 inch x 0.06 inch (I.D. × O.D.) tubing (Tygon® ND-100-80, autoclaved before usage), and then the HFSM was placed in the cell culture incubator. Cell culture media (RPMI-1640 containing 10% FBS and 1% PS) in the HFSM was changed every 12 hr.

## Flow simulation

To predict the type of flow in the HFSM, the concentration profiles of the sample fluid were simulated in COMSOL Multiphysics 5.5 (COMSOL Inc.) with a 2D laminar flow model. Since all fluids used in this study were aqueous solutions, we considered all sample and side solutions as water with a density of 997.1 kg/m$^3$ and a dynamic viscosity of 8.91×10$^{-4}$ Pa•s at room temperature (300K).

### Treating, collecting, and imaging cells in the HFSM

After culturing 4T1 cells in the HFSM for 7 days, the device was moved into a stage top incubator with a confocal fluorescence microscope (Zeiss, LSM 710) for maintaining cell viability during subsequent treatment and imaging. The syringes containing sample and buffer solutions were positioned on the pumps and the tubing (Tygon® ND-100-80 (0.02 inch x 0.06 inch)) was connected to the syringe needle and the other end was plugged into the hole of the PMDS layer of the device. Different sample solutions including the green food coloring dye (ESCO foods Inc.), DiI (DiIC18(3), 5 μM), doxorubicin (2 mg/ml in 0.9% saline containing 0.02% Triton X-100), and trypsin EDTA (0.25% w/v) were prepared. DiI stock solution was prepared by dissolving DiI in DMSO at 5 mM and stored in the dark at 4˚C. Right before use, 5 μL stock solution was diluted in 5 mL cell culture media to make 5 μM staining solution. The sample solution and cell culture media were pumped into the sample inlet and two side inlets, respectively, using three digital syringe pumps and three 5 ml syringes (BD Luer-Lok™, sterile). All syringes were wrapped with heating pads (New Era Inc.) to keep the temperature of solutions at 37˚C. The path of the sample fluid stream was controlled by changing the flow rates of cell culture media in the two side channels. The cells were collected through the outlet after exposure to trypsin EDTA flow. Brightfield and fluorescence images of cells in the HFSM were acquired with the microscope using low- (Zeiss A-Plan, 5x, dry and Zeiss Plan-Apochromat, 10x and 20x, dry) and high- (Zeiss Plan-Apochromat, 63x, oil) magnification objectives.

## Results and discussion

### Simulation and experimental testing of laminar flow in the HFSM

The width ($W_f$) and path of a hydrodynamically focused sample stream depend on the following dimensional parameters: (1) the flow rate of sample fluid $Q_s$ and the flow rate of the side fluid $Q_{si}$, and (2) the density ρ and dynamic viscosity μ of the fluids. Assuming sample and side fluids have similar physical properties with water, the Reynolds number Re = $\rho Q_{cc} D/\mu$, where $Q_{cc}$ is the flow rate of fluid in the culture chamber and D is the characteristic length of the culture chamber, was less than 1 based on our calculation when $Q_{cc}<0.01$. Thus, hydrodynamic focusing of a sample fluid in our device is governed mainly by the flow rates of sample and side fluids and minimal diffusion occurs as shown in our model and experiments.

To optimize and characterize the microfluidic device design, we performed simulations of the hydrodynamic focusing and shifting of sample fluid with a 2D laminar flow model (S1 and S2 Figs and Fig 1B). Sample fluid ($H_2O$ having 1 arbitrary concentration unit (a.u.)) was hydrodynamically focused along the center channel by two side fluids ($H_2O$ having 0 a.u.). To simulate the behavior of the sample stream with different flow rate ratios, we applied a range of values for the fluid velocity of the main (0.001–0.0025 m/s corresponding to a flow rate range of 2–5 μL/min) and side (0.0025–0.0175 m/s corresponding to a flow rate range of 5–35 μL/min) inlets. Pseudocolor images of the simulation data showed laminar flows and concentration distributions of the sample fluid with different inlet flow rate ratios. With unequal flow rates of the two side fluids $Q_{si}$, the path of the focused sample stream could be shifted left and right from the center of the main channel. With these simulation conditions, we evaluated HFSM device designs with three different inlet geometries: 1) cross-channel with an inverted T-shape junction, 2) 45˚ tilted side channel with a down arrow-shaped junction, and 3) sample/side co-flow channel without a junction (S1 Fig). The simulation data show that the third configuration of sample/side co-flow input channels was the optimal device inlet geometry for creating the narrowest width and precisely controlled path of a hydrodynamically focused and shifted sample stream. We determined the mean width of the focused stream by

measuring the full width at half maximum (FWHM) of the concentration profiles. To change the position of the sample stream, we turned off the input pump for the sample fluid, adjusted the flow rates of the left and right side fluids to preset the new position of the sample stream before turning on the input pump for the sample fluid. This pump off-on method can prevent unselected cells from exposure to sample molecule during repositioning of sample solution stream. Therefore, simulation of the time-dependent concentration change of the shifted sample stream right after turning on its input pump is necessary to characterize hydrodynamic shifting in the co-flow inlet device. Under the preset side flow rates for shifting the position of the sample stream at a flow rate ratio of 35:2:5 (µL/min), it took 5 seconds for the shifted sample stream to reach its peak concentration across the entrance of the culture chamber after turning on the input pump for the sample fluid. The mean width (264 ± 4 (SD) µm) of the sample stream did not significantly change during this time period (S2 Fig). In addition to shifting the sample stream in the optimal device design at flow rate ratios of 5:2:35, 20:2:20, and 35:2:5 (µL/min) in Fig 1B, we simulated an increase in the flow rate of the sample fluid while keeping the same flow rates for both side fluids (i.e., 20:2:20 to 20:5:20 (µL/min)), as shown in the 2nd and 4th images in Fig 1B.

To confirm the simulation data of flow focusing and shifting in Fig 1B, we conducted experiments under similar conditions. Green food coloring dye and phosphate buffered saline (PBS) were pumped into the central inlet and side inlets of the HFSM, respectively, at different flow rate ratios using digital syringe pumps (NE-300 New Era). The range of inlet flow rates for the food dye solution was from 2 to 5 µL/min while the range for PBS was from 5 to 35 µL/min. Fig 1C shows time-averaged snapshots of the food dye stream in the main channel of the HFSM acquired in brightfield imaging mode by confocal fluorescence microscopy. PBS from the left and right side inlets squeezed the food dye stream from the main inlet as a co-flow-focusing. The food dye flow formed a laminar stream with a narrow width that continued down to the culture chamber. The shift in the food dye stream due to uneven side inlet velocities of PBS is also apparent.

We further analyzed the lateral concentration of the sample stream at the entrance to the culture chamber (indicated by red dotted lines in Fig 1A and 1B (left)) through simulation profiles (Fig 2A). The mean width of the focused streams at flow ratios of 20:2:20 and 20:5:20 (µL/min) is 488 µm in both cases. The simulated peak concentration of the sample fluid in the culture chamber increased 1.8 times (from 0.2 to 0.37 a.u.) as the inlet flow rate of the sample fluid $Q_s$ rose from 2 to 5 µL/min.

Fig 2B shows the strong correlation between simulation predictions and experimental measurements of shifting the sample stream under different flow rate ratios. The distance was measured from the centerline of the culture chamber. A distance of 0 indicates that the sample stream is directly along the center of the culture chamber while negative and positive values represent left and right shifts of the sample stream, respectively. The sample stream could be shifted in ±0.5 mm lateral range from the center of culture chamber. These simulation and experimental results demonstrate the capability for precise microfluidic flow control for spatially selective delivery of drugs to cells.

## Application of the HFSM for spatially controlled, cell-based drug testing

To demonstrate the feasibility of our device to spatially select cells for treatment prior to collection for an integrative cell-based drug assay, we treated a mouse mammary tumor cells (4T1) with a lipophilic fluorophore (DiI), a chemotherapeutic agent (doxorubicin), and cell detachment enzyme (Trypsin EDTA) solutions as sample fluids. High- and low-magnification bright-field and fluorescence microscopy was used as the primary monitoring method.

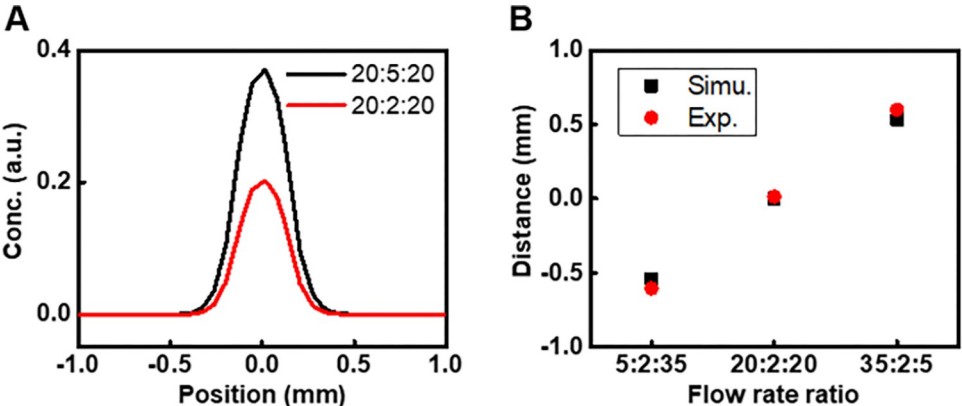

**Fig 2. Quantitative profiles of hydrodynamic focusing and shifting in the HFSM. A** Simulation profiles of differences in lateral concentration of sample fluid at the entrance to the culture chamber in the device for different sample flow rates. **B** Simulation predictions and experimental measurement of distance of the sample stream from the centerline of the culture chamber at different side flow rates.

## Culturing tumor cells in the HFSM

After being loaded with 4T1 cells, the HFSM was moved into a humidified cell incubator. 4T1 cells grew and covered more than 75% of the cell culture chamber area after incubation for 7 days at 37˚C with 5% $CO_2$ in a humidified atmosphere, as shown in Fig 3. This result demonstrates our device can provide proper environmental conditions for culturing and housing millions of adherent 4T1 tumor cells to examine spatially selective cell treatment and collection.

## Spatially selective cell staining with lipophilic fluorescent dye

The HFSM device containing 4T1 cells was placed on the stage of a confocal fluorescence microscope (Zeiss, LSM 710) and a stage top incubator covered the device to maintain cell viability during imaging. A DiI staining solution and cell culture media were pumped into the sample channel (center) and two side channels (left and right), respectively, using digital syringe pumps. The path of the DiI stream in the culture chamber could be controlled by the flow rates of the cell culture media in the side inlets. By confocal fluorescence microscopy with a 514 nm excitation laser, 519–673 nm emission filter, and 5x dry objective, 4T1 cells were visualized by DiI dye staining (Fig 4A). Three vertical stripes of fluorescently stained 4T1 cells were marked in the culture chamber by focusing and shifting the DiI stream with flow rate ratios of 5:2:35, 20:2:20, and 35:2:5 (µL/min). The flow time for each stream was 5 min. Fig 4B shows the strong fluorescence signal of DiI in the plasma member and cytoplasm of 4T1 cells. Thus, our device created hydrodynamic flow focusing and shifting that was successfully applied for spatially selective delivery of a reagent to cells in the 2 mm-wide culture chamber.

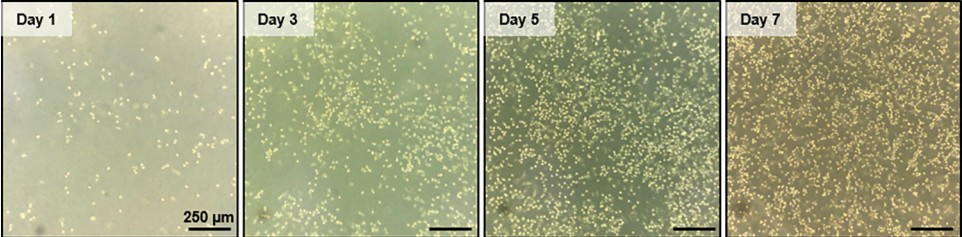

**Fig 3. Proliferation and growth of 4T1 tumor cells in the HFSM device.** Brightfield microscope images of 4T1 cells in the culture chamber over the course of a 7-day incubation.

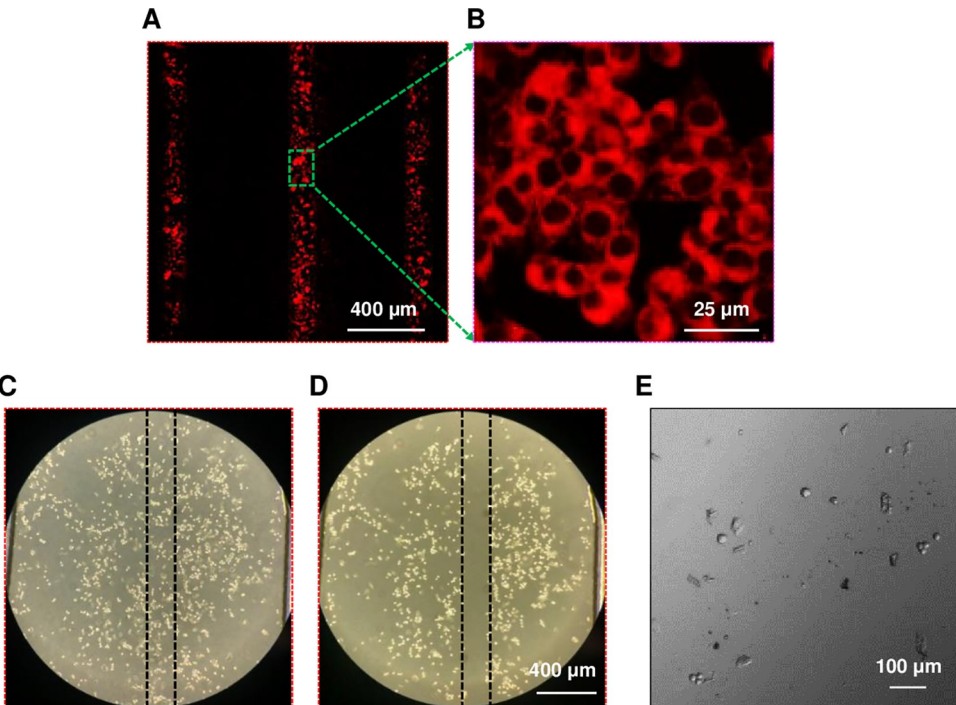

**Fig 4. Spatially controlled staining and collection of 4T1 tumor cells using the HFSM. A** A single fluorescence image of 4T1 cells visualized after staining by three different DiI stream paths (flow rate ratios of 5:2:35, 20:2:20, and 35:2:5 (μL/min)). **B** High magnification image of 4T1 cells stained with DiI at a flow rate ratio of 20:2:20 (μL/min). **C, D** Brightfield images of 4T1 cells in the culture chamber (**C**) before and (**D**) after exposure to trypsin EDTA flow. Black dotted lines present the stream path of trypsin EDTA. **E** Brightfield image of collected 4T1 cells after detachment from the chamber by exposure to trypsin EDTA flow.

These results demonstrate the potential for testing different drug compounds or drug doses for millions of cells in a single device by spatially controlled delivery.

## Spatially selective cell collection with trypsin EDTA flow

4T1 cells were selectively collected from the HFSM device by perfusion of 0.25% (w/v) trypsin EDTA. After loading and culturing 4T1 cells in a fresh HFSM device, the device was moved onto the microscope and contained in a stage top incubator. Trypsin-EDTA solution and cell culture media were constantly pumped into the sample inlet and two side inlets, respectively. The flow rates for trypsin EDTA solution and cell culture media were 5 μL/min and 20 μL/min, respectively. 4T1 cells attached to the bottom of the culture chamber before trypsin EDTA flow passes over (Fig 4C). 4T1 cells under the path of trypsin EDTA flow (area between black dashed lines in Fig 4D) detached and washed out of the HFSM device after 40 min exposure. Using ImageJ, we determined that the cell area of 79% in between the black dashed lines in the Fig 4C and 4D was disappeared after trypsin EDTA flow treatment (S3 Fig). The suspension of detached cells was collected through the outlet of the device (Fig 4E). These results demonstrate a proof of concept for our device and overall approach for spatially selective cell collection for downstream single-cell and molecular analyses.

## Spatially selective cell treatment with a chemotherapy drug

Doxorubicin is a cytotoxic molecule currently used for chemotherapy of breast and ovarian cancers [22–24]. We tested the ability of our device to spatially deliver doxorubicin to cells and

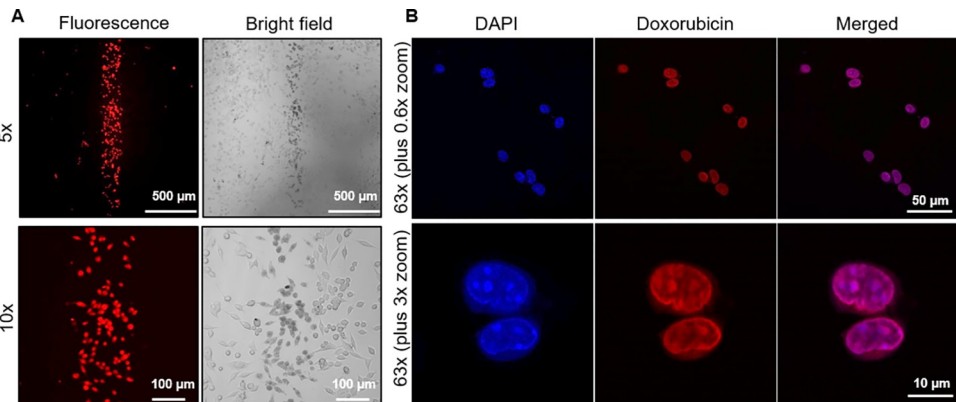

**Fig 5. Spatially selective treatment and collection of 4T1 tumor cells for testing subcellular doxorubicin delivery using the HFSM. A** Low-resolution fluorescence (right) and brightfield (left) images of 4T1 cells after exposure to a stream of doxorubicin for 10 min. Images at the top and bottom were obtained using 5x and 10x dry objectives, respectively. The red fluorescence signal represents doxorubicin internalized in 4T1 cells. **B** High-resolution fluorescence images of a 4T1 cell suspension collected using spatially controlled trypsin EDTA flow post-exposure to doxorubicin flow. The top and bottom images were acquired using a 63x oil objective and zoom functionality (0.6x and 3x zoom), respectively. The high-resolution images show clear co-localization of the fluorescence signals of DAPI (blue) and doxorubicin (red) in the cell nuclei, verifying nuclear delivery of doxorubicin using the HFSM.

selectively collect the treated cells. 4T1 cells were loaded and cultured in a HFSM device for 7 days as described above. Again, the HFSM device loaded with 4T1 cells was placed on the stage of the confocal fluorescence microscope and contained within a stage top incubator. Doxorubicin solution (2 mg/ml in 0.9% saline, containing 0.02% Triton X-100) and cell culture media were constantly pumped into the sample inlet and two side inlets at flow rates of 2 μL/min and 20 μL/min, respectively. Fig 5A shows the fluorescence and brightfield contrast images of 4T1 cells exposed to the doxorubicin stream for 10 min. Doxorubicin molecules have intrinsic fluorescence [25, 26]. Thus, the doxorubicin internalized in 4T1 cells was able to be visualized with a 488 nm excitation laser and 535–674 nm emission filter using 5x and 10x dry objectives (top and bottom, Fig 5A). Doxorubicin can induce therapeutic effects (i.e. cytotoxicity) to a cancer cell only when the drug molecules interact with DNA in the cell nucleus. However, while low magnification objectives having long working distances (9.9 mm and 2 mm for 5x and 10x objectives, respectively) can image many cells at once across a 1 mm thick glass slide at the bottom of the device, the same objectives have a low numerical aperture (NA) of 0.12 and 0.45, respectively, and cannot provide subcellular resolution images that clearly show cell nuclei. Thus, to confirm nuclear delivery of doxorubicin, we collected the treated 4T1 cells after exposure to trypsin EDTA flow and applied high-resolution imaging of the cell suspension using a 63x oil objective (working distance = 0.19 mm, NA = 1.4) (Fig 5B). By counterstaining DNA with 4',6-diamidino-2-phenylindole (DAPI), high-resolution images clearly show strong co-localized signal of doxorubicin in the cell nuclei. This downstream analysis offered further evaluation of subcellular targeting of the drug compound. Other assays, such as flow cytometry and single-cell sequencing, might also be applied to single cell suspensions, offering more detailed information on drug treatment effects at the cellular and molecular levels.

Given the result above, we sought to determine the cytotoxic effect of doxorubicin on the tumor cells. We selectively delivered doxorubicin to 4T1 cells (at a flow rate ratio of 20:2:20 (μL/min), 10 min duration) in a fresh HFSM device as described above, placed the device in a cell culture incubator for 12 hr, and changed the media. Fig 6 shows the cytotoxic effect that doxorubicin had on the 4T1 cells that were selectively exposed to the drug flow. The 4T1 cells died and washed out from the device after the 12 hr incubation.

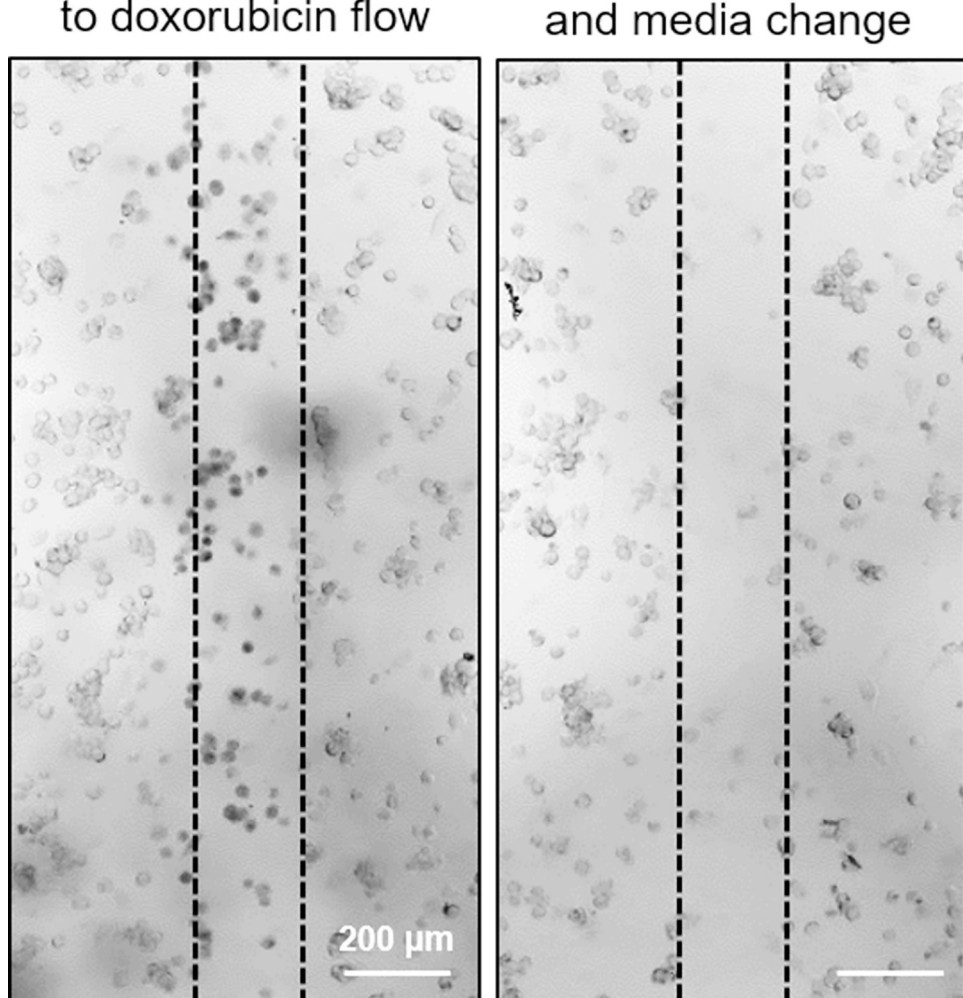

**Fig 6. Cytotoxicity assay of doxorubicin on spatially selected 4T1 cells in the HFSM device.** Brightfield microscope images of 4T1 cells in the culture chamber immediately after a 10 min exposure to a doxorubicin stream (left) and after a subsequent 12 hr incubation and one time media change (right). The area between the black dashed lines indicates the path of the doxorubicin stream.

## Conclusion

In this study, we demonstrated hydrodynamic focusing and shifting as an advanced microfluidics approach for cell-based drug testing. Our optimized microfluidics device can readily control the path of a drug or reagent solution over the culture chamber for spatially selective delivery to cells. It is also capable of growing millions of cells by providing a large culture chamber (2 mm width). These technical advantages permitted not only *in situ* monitoring of drug effects but also selective cell collection for downstream high-resolution analysis. Although it requires further studies and optimization in cell culturing and spatially selective collection to expand the application of the device for other cell types, we anticipate that this HFSM approach and device can be applied broadly to facilitate the simultaneous evaluation of cell responses to various drug treatments under fluid dynamic conditions mimicking physiological blood flow. In particular, through integration with other assay methods, the HFSM will

provide comprehensive information on the effects of drug candidates at the cellular and molecular levels for drug screening and discovery research.

## Supporting information

**S1 Fig. Simulation of hydrodynamic flow focusing and shifting in different HFSM designs.** (TIF)

**S2 Fig. Simulation of the time-dependent concentration changes of a hydrodynamically shifted sample flow.** (TIF)

**S3 Fig. Reduction of cell area on the stream path of trypsin EDTA solution.** (TIF)

## Acknowledgments

We thank Dr. Stephen J. Kron at the University of Chicago for providing 4T1 cell line. We also thank the Fluorescence Imaging Core at the University Illinois Chicago for their technical support in confocal fluorescence microscopy and Evan H. Phillips for manuscript editing assistance.

## Author Contributions

**Conceptualization:** Xu Wang, David T. Eddington, Steve Seung-Young Lee.

**Data curation:** Xu Wang.

**Formal analysis:** Xu Wang, Steve Seung-Young Lee.

**Funding acquisition:** Steve Seung-Young Lee.

**Investigation:** Xu Wang, Jingtian Zheng.

**Methodology:** Xu Wang, Maheshwar Adiraj Iyer, Adam Henry Szmelter.

**Supervision:** David T. Eddington, Steve Seung-Young Lee.

**Writing – original draft:** Xu Wang, Steve Seung-Young Lee.

**Writing – review & editing:** Xu Wang, Jingtian Zheng, David T. Eddington, Steve Seung-Young Lee.

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
