## [Decision Letter · Decision Letter 0]

22 Jul 2022

PONE-D-22-11631Spatially selective cell treatment and collection for integrative drug testing using hydrodynamic flow focusing and shiftingPLOS ONE

Dear Dr. Lee,

Thank you for submitting your manuscript to PLOS ONE. After careful consideration, we feel that it has merit but does not fully meet PLOS ONE’s publication criteria as it currently stands. Therefore, we invite you to submit a revised version of the manuscript that addresses the points raised during the review process.

We look forward to receiving your revised manuscript.

Kind regards,

Giuseppe Chirico

Academic Editor

PLOS ONE

Journal Requirements:

“This study was supported by NIBIB R00 EB0022636 and NIGMS R35 GM142743 to S. S.-Y. Lee. Experiments were performed in a renovated laboratory space at University of Illinois Department of Pharmaceutical Sciences, supported by NIH C06 RR015482.”

4. Please expand the acronym “NIBIB, NIGMS and NIH” (as indicated in your financial disclosure) so that it states the name of your funders in full.

Additional Editor Comments:

Basing on my personal reading of the Ms. and the review reports I suggest the Authors to review their Ms. and resubmit it. In particular, I would like to stress the following points:

(1) Both the Reviewers ask to expand the literature analysis. Similar microfluidic experiments are reported. It is very much desired that the reader can understand what are the technical advances in this paper and the differences with the published studies.

(2) Please, provide a characterization of the switching/stabilization of flow positions.

(3) please, outline the advantages of the presented device over conventional cell culture systems.

I would appreciate to have marked changes in the revised submission and a detailed comments to the points highlighted by the Reviewers.

thank you

Reviewers' comments:

Reviewer's Responses to Questions

**Comments to the Author**

1. Is the manuscript technically sound, and do the data support the conclusions?

Reviewer #1: No

Reviewer #2: Yes

2. Has the statistical analysis been performed appropriately and rigorously? 

Reviewer #1: No

Reviewer #2: N/A

3. Have the authors made all data underlying the findings in their manuscript fully available?

Reviewer #1: Yes

Reviewer #2: Yes

4. Is the manuscript presented in an intelligible fashion and written in standard English?

Reviewer #1: Yes

Reviewer #2: Yes

5. Review Comments to the Author

Reviewer #1: Wang et al. reported a microfluidic device for treating cells with drugs using flow-focusing and switching methods. The authors cultured mammalian cells at the bottom of the flow channel and then treated the cells with fluorescent dyes and antitumor drugs. They also used a trypsin/EDTA solution to locally detach the cells and collect them. Although the manuscript is well organized, I do not recommend this paper due to some critical weaknesses. Details are as follows.

(1) First of all, similar microfluidic experiments were performed 10-20 years ago. The authors have not verified the technical advances in this paper, so it lacks novelty.

(2) The authors do not properly characterize the switching behavior of the flow position. In a laminar flow system such as the one in this paper, switching/stabilization of flow positions takes a certain amount of time, which is critically associated with the accuracy of cell processing. However, there is no description on this point.

(3) The system presented is not practical as a platform for evaluating the effects of drugs on cells. The advantages over conventional cell culture systems, including 96-well plates, are not described.

Reviewer #2: The Authors present a simple microfluidic device with a wide cell culture chamber and demonstrate the use of hydrodynamic focusing and shifting to selectively expose cells to treatments and to retrieve cells via selective exposure to Trypsin.

The device has many potential applications. Experimental results are supported by modelling and simulations. The paper is clear and well organized. Figures are carefully designed.

I have only one suggestion, to better highlight the novelty of this work. I think the Authors should expand the literature survey at lines 51-57, providing more details about existing devices and approaches. In this way, the statement at lines 58-61 would be better motivated and supported.

6. PLOS authors have the option to publish the peer review history of their article (what does this mean?). If published, this will include your full peer review and any attached files.

Reviewer #1: No

Reviewer #2: No

---

## [Author Response · Author response to Decision Letter 0]

2 Sep 2022

Please find our response to Editor's and Reviewers' comments in the submitted 'Cover Letter' and ' Response to Reviewers' documents.

---

## [Decision Letter · Decision Letter 1]

9 Nov 2022

PONE-D-22-11631R1Spatially selective cell treatment and collection for integrative drug testing using hydrodynamic flow focusing and shiftingPLOS ONE

Dear Dr. Lee,

Thank you for submitting your manuscript to PLOS ONE. After careful consideration, we feel that it has merit but does not fully meet PLOS ONE’s publication criteria as it currently stands. Therefore, we invite you to submit a revised version of the manuscript that addresses the points raised during the review process.

As you see from the revisions, one additional reviewer was contacted by me and raised a few additional minor points that I suggest to address in a further revision of you Ms. You find the details in the review reports. After these clarifications on some experimental procedures, the Ms. can accepted for publication.  Please submit your revised manuscript by Dec 24 2022 11:59PM. If you will need more time than this to complete your revisions, please reply to this message or contact the journal office at plosone@plos.org. Please include the following items when submitting your revised manuscript:A rebuttal letter that responds to each point raised by the academic editor and reviewer(s). You should upload this letter as a separate file labeled 'Response to Reviewers'.A marked-up copy of your manuscript that highlights changes made to the original version. You should upload this as a separate file labeled 'Revised Manuscript with Track Changes'.An unmarked version of your revised paper without tracked changes. You should upload this as a separate file labeled 'Manuscript'.If applicable, we recommend that you deposit your laboratory protocols in protocols.io to enhance the reproducibility of your results. Protocols.io assigns your protocol its own identifier (DOI) so that it can be cited independently in the future. For instructions see: https://journals.plos.org/plosone/s/submission-guidelines#loc-laboratory-protocols. Additionally, PLOS ONE offers an option for publishing peer-reviewed Lab Protocol articles, which describe protocols hosted on protocols.io. Read more information on sharing protocols at https://plos.org/protocols?utm_medium=editorial-email&utm_source=authorletters&utm_campaign=protocols.

We look forward to receiving your revised manuscript.

Kind regards,

Giuseppe Chirico

Academic Editor

PLOS ONE

Journal Requirements:

Additional Editor Comments:

I thank you for the revisions of the original Ms. that is now in a form that can be accepted for publication after you address the comments of reviewer #2. In particular the clarification of some experimental results and discussions that appear to be still are unclear.

In addition to this, a short description on the limitation of this device and method would also improve the discussion.

Reviewers' comments:

Reviewer's Responses to Questions

**Comments to the Author**

1. If the authors have adequately addressed your comments raised in a previous round of review and you feel that this manuscript is now acceptable for publication, you may indicate that here to bypass the “Comments to the Author” section, enter your conflict of interest statement in the “Confidential to Editor” section, and submit your "Accept" recommendation.

Reviewer #1: All comments have been addressed

Reviewer #3: (No Response)

2. Is the manuscript technically sound, and do the data support the conclusions?

Reviewer #1: Partly

Reviewer #3: Partly

3. Has the statistical analysis been performed appropriately and rigorously? 

Reviewer #1: Yes

Reviewer #3: Yes

4. Have the authors made all data underlying the findings in their manuscript fully available?

Reviewer #1: Yes

Reviewer #3: Yes

5. Is the manuscript presented in an intelligible fashion and written in standard English?

Reviewer #1: Yes

Reviewer #3: Yes

6. Review Comments to the Author

Reviewer #1: After revision, the author has adequately answered my previous questions/comments. I still question the novelty of this paper, but I recommend that this paper be accepted for publication.

Reviewer #3: This paper reports the development of a microfluidic device for drug testing based on controlling hydrodynamics. The authors designed and simulated three microfluidic devices suitable for creating the sample flow stream. After optimizing the microfluidic device, they demonstrated the cell culture, selective cell staining, selective cell collection, and selective cell treatment with doxorubicin. The microfluidic device proposed in this paper could perform selective drug delivery and cell-based drug testing. The experimental results well supported the concept of the paper. However, several experimental results and discussions are unclear. Therefore, I recommend this paper for publication in PLOS ONE after minor revision.

Comments

1. How did they connect between the syringe pumps and syringes? Please provide information about the tubing (or connection to the microfluidic device and syringes).

2. Did they treat the microchannel surface to adhere cells onto the microchannel? For the cell culture in the microfluidic device, how did they prevent drying out the cell culture medium? PDMS shows high gas permeability, and the microchannel may be dried out during the incubation. In addition, what is the confluency of the cells before each experiment?

3. On page 10, line 226, how did they measure the flow width?

4. Why did they turn off the syringe pump for the sample stream? Generally, the flow rate could be changed without turning off the syringe pump.

5. On page 12, line 275, please elaborate the differences in growth rate compared with the typical cell culture method and other on-chip cell culture methods.

6. For the cell stain experiments, is there any effect of molecular (DiI) diffusion? In addition, when the flow rate was changed to shift the DiI stream, the DiD stream could stain undesirable cells onto the microchannel due to the shift of the flow stream. Please comment on it and how to prevent undesirable cell staining.

7. What is the cell recovery rate for the selective cell collection experiment?

8. For the selective cell collection and cytotoxic effect of doxorubicin, please provide the cell viability.

9. Please discuss the limitation of this device and method. Is there a possibility of cell type effects?

7. PLOS authors have the option to publish the peer review history of their article (what does this mean?). If published, this will include your full peer review and any attached files.

Reviewer #1: No

Reviewer #3: No

---

## [Author Response · Author response to Decision Letter 1]

22 Nov 2022

Please find our response in the 'Response to Reviewers' file.

---

## [Editor Report · Decision Letter 2]

1 Dec 2022

Spatially selective cell treatment and collection for integrative drug testing using hydrodynamic flow focusing and shifting

PONE-D-22-11631R2

Dear Dr. Lee,

We’re pleased to inform you that your manuscript has been judged scientifically suitable for publication and will be formally accepted for publication once it meets all outstanding technical requirements.

Kind regards,

Giuseppe Chirico

Academic Editor

PLOS ONE
---

## [Editor Report · Acceptance letter]

5 Dec 2022

PONE-D-22-11631R2 

Spatially selective cell treatment and collection for integrative drug testing using hydrodynamic flow focusing and shifting 

Dear Dr. Lee:

I'm pleased to inform you that your manuscript has been deemed suitable for publication in PLOS ONE. Congratulations! Your manuscript is now with our production department. 

Kind regards, 

on behalf of

Dr. Giuseppe Chirico 

Academic Editor

PLOS ONE